# Self-Adaptive Flask-like Nanomotors Based on Fe_3_O_4_ Nanoparticles to a Physiological pH

**DOI:** 10.3390/nano12122049

**Published:** 2022-06-15

**Authors:** Tianyu Gao, Jinwei Lin, Leilei Xu, Jianguo Guan

**Affiliations:** State Key Laboratory of Advanced Technology for Materials Synthesis and Processing, School of Materials Science and Engineering, Wuhan University of Technology, Wuhan 430070, China; gty@whut.edu.cn (T.G.); linjinwei@whut.edu.cn (J.L.)

**Keywords:** nanomotors, pH responsiveness, Fe_3_O_4_ nanoparticles, dual enzyme-like activities, physiological pH chemotaxis

## Abstract

In living bodies, pH values, which are precisely regulated and closely associated with diseased cells, can act as an efficient biologically intrinsic indicator for future intelligent biomedicine microsystems. In this work, we have developed flask-like carbonaceous nanomotors (FCNMs), via loading Fe_3_O_4_ nanoparticles (NPs) into a cavity, which exhibit a self-adaptive feature to a specific physiological pH by virtue of the pH-dependent dual enzyme-like activities of Fe_3_O_4_ NPs. Specifically, the peroxidase-like activity of Fe_3_O_4_ NPs in an acidic pH range, and the catalase-like activity in a near neutral and alkaline pH range, determine the products in the motion system (•OH, ions and O_2_), whose diffusions from the inner to the outside of the flask result in fluid movement providing the driving force for the movement of the FCNMs. Correspondingly, changes of the product concentrations and species in the physiological pH range (4.4–7.4) result, firstly, in velocity decrease and, then, with increase in pH, increase of the FCNMs occurs. Thanks to the non-linear velocity responsiveness, the FCNMs show intriguing pH taxis towards 6.8 (generally corresponding to the physiological pH in tumor microenvironments), where a maximum velocity appears. Furthermore, the superparamagnetic feature of the Fe_3_O_4_ NPs simultaneously endows the FCNMs with the abilities to be magnetic-oriented and easily separated. This work could significantly increase the possibility of nanomotors for targeted therapy of tumors and next-generation biotechnological applications.

## 1. Introduction

Micro/nanomotors (MNMs), inspired by natural microorganisms, are artificial power devices between nano- and micro-scale that can convert energy in the environment into efficient motion [1]. Their excellent movement characteristics, especially when moving in micro-scale regions or enclosed areas, and chemically rich features show great promise for the revolutionization of biomedical applications [2,3,4,5,6,7,8,9,10,11].

Smart drug delivery with active targeting abilities is essential for futuristic biomedical fields. Compared with external field-propelled MNMs, chemically driven MNMs can harvest energy from their surroundings to induce movement, which provides a possibility for responding to local environments. In recent years, a growing effort has been devoted to engineer diverse chemically driven MNMs to respond to various stimuli, including metabolite concentration [12,13], microenvironment temperature [14,15], etc. In living bodies, pH value is an efficient biologically intrinsic indicator because it is highly diverse and precisely regulated in different organs and tissues. For example, the extracellular pH for healthy human tissues varies around 7.4, whereas the pH in tumor microenvironments is marginally acidic, ranging from 6.5 to 7.0 (average ~6.8), which further drops to 4.4–6.0 within tumor cells [16]. Minor changes of pH values in specific areas may affect physiological metabolism and even lead to diseases. Therefore, the various pH values can act as chemical signals to induce the responsive motion of MNMs. For example, the pH-responsive MNMs fabricated by catalytic materials, such as bio-enzymes [17], nano-enzymes [18,19,20,21] or active metals [22,23,24,25,26], the catalytic activities of which are influenced by environmental pH, usually show change of velocities and directions upon regulating the “fast/slow” aspect of pH-dependent driving reactions. Introducing pH-responsive polymers into MNMs, as gatekeepers to regulate the accessibility of “fuel”, also easily achieve “on/off” of MNMs [21,27]. Additionally, pH-induced hydrophilicity/hydrophobicity transition and surface tension variation are also used to control the motion of MNMs displaying pH-responsiveness [28,29]. Although some well-designed MNMs can display a responsive motion behavior using pH as an indicator, the responsiveness levels are only indicated by monotonous velocity changes or transient on/off. However, besides simply identifying their microenvironments, the self-adaptive movement of MNMs to a specific pH, like some microorganisms with chemotaxis that are able to autonomously move to specific areas, is indispensable for futuristically smart motor systems. It means that they are able to seek focal sites in terms of a specific pH actively, and thus realize targeted drug delivery toward tumors.

Herein, we designed and constructed flask-like carbonaceous nanomotors (FCNMs) that exhibit self-adaptive pH-responsive motion behavior in the physiological pH range (4.4–7.4) on the basis of pH-dependent dual enzyme-like activities of Fe_3_O_4_ NPs. The non-monotonic change of velocities for as-prepared FCNMs are influenced by the diffusion behaviors of pH-dependent products. At a lower pH range (4.4–5.6), Fe_3_O_4_ NPs show peroxidase-like activity and the FCNMs are propelled by the diffusion of the product •OH and ions from the buffer solution used to balance the pH value of the system. With increase of the pH values, the peroxidase-like activity decreases until it disappears, while the catalase-like activity of Fe_3_O_4_ NPs appears and increases. This leads to the transformation of the driving mechanisms from ionic and non-ionic diffusiophoresis to non-ionic diffusiophoresis. Correspondingly, the as-prepared FCNMs show a maximum motion velocity of 12.4 body lengths s^−1^ at pH 6.8 within 20 mM buffer solution and a fantastically spontaneous pH-chemotaxis from 7.4 to 6.8, which has a tempting application prospect for targeting tumor microenvironments. Additionally, thanks to the superparamagnetism of Fe_3_O_4_ NPs, long-range motion control of the FCNMs is easily achieved by a magnetic field. These findings provide some insights into the pH-responsive motion mechanisms of Fe_3_O_4_-based MNMs, and offer some possibilities for application in biomedical fields, especially in the targeted therapy of tumors.

## 2. Materials and Methods

### 2.1. Materials and Reagents

Iron chloride (FeCl_3_·6H_2_O, ≥98%), oleic acid (OA, 85%), dimethylsulfoxide (DMSO, ≥99.5%), triethylamine (≥99%), acetone, n-hexane, chloroform and ethanol were purchased from TCI (Shanghai, China) Development Co., Ltd., China. Oleyl alcohol (80~85%) was purchased from Weng Jiang Reagent Co., Ltd. (Guangdong, China). Sodium oleate (C_17_H_33_CO_2_Na, >97%), dimercaptosuccinic acid (DMSA, 98%) and diphenyl ether (>99%) were purchased from Aladdin Biochemistry Co., Ltd. (Shanghai, China). D-ribose (≥98%) and poly(ethylene glycol)-block-poly(propylene glycol)-block-poly(ethylene glycol) (EO_20_-PO_70_-EO_20_, P123) were purchased from Sigma-Aldrich (Shanghai, China). Deionized water was produced in a Milli-Q system (Millipore, MA, USA) for the preparation of all aqueous solutions.

### 2.2. Synthesis of Fe_3_O_4_@OA NPs

The Fe_3_O_4_@OA NPs were synthesized according to previously reported procedure, with a slight modification [30]. Briefly, 3.6 g of Fe(oleate)_3_ precursor, 1.14 g of oleic acid, and 3.22 g of oleyl alcohol were dissolved in 20 g of diphenyl ether at room temperature. The mixture was first heated to 90 °C at a constant heating rate of 8 °C/min in a nitrogen atmosphere, kept for 2 h to remove dissolved oxygen, and then heated to 220 °C to maintain this temperature for 10 min. After the reaction, the mixed solution was quickly cooled to room temperature, and the nanoparticles were washed three times with a mixed solvent of acetone and n-hexane by centrifugal separation, and finally dispersed in chloroform.

### 2.3. Modification of Fe_3_O_4_@OA NPs by DMSA

Fe_3_O_4_@DMSA NPs was prepared according to the previously reported method [31]. Briefly, 100 mg Fe_3_O_4_@OA NPs were first dispersed in 10 mL chloroform and 50 μL triethylamine were added. Next, 50 mg DMSA was dissolved in 10 mL DMSO and added to the above solution. The resulting mixed solution was magnetically stirred vigorously for 12 h at 60 °C. Then, the nanoparticles were washed three times with ethanol by centrifugal separation. In order to modify more DMSA on the surface of the Fe_3_O_4_ NPs, and improve the water solubility, the nanoparticles were once again dispersed in ethanol for surface double-exchange. After repeating the previous operation, the obtained Fe_3_O_4_@DMSA NPs were washed with deionized water and dispersed in it.

### 2.4. Synthesis of Flask-like Carbonaceous Carriers (FCCs)

FCCs were prepared according to the previously reported method [32,33]. Briefly, 0.0365 g of sodium oleate and 0.0435 g of P123 were dissolved in 20 mL deionized water and stirred slowly for a period of time at 40 °C until it became clarified. Next, 40 mL aqueous solution, containing 3 g D-ribose, was added to the above solution. The mixed solution was transferred to a 100 mL autoclave and hydrothermally treated at 160 °C for 12 h. Then, the resulting FCCs were washed with deionized water three times.

### 2.5. Preparation of FCNMs

A certain number of dry FCCs were added to the water/ethanol mixed solution containing Fe_3_O_4_@DMSA NPs. Next, the mixture was treated in an ultrasonic instrument for 20 min to allow the FCCs to be fully dispersed and the Fe_3_O_4_@DMSA NPs to enter the hollow structure. The excess Fe_3_O_4_@DMSA NPs were removed by repeated washing and centrifugation until the supernatant was transparent. Then, the obtained products were placed in a vacuum drying oven at 60 °C for 12 h. The procedure was repeated three times and, finally, the resulting FCNMs were scattered in the deionized water.

### 2.6. Observation of the Motion of the FCNMs

The motion of the as-prepared FCNMs was observed by using a Leica DM 3000B optical microscope with a high-resolution CCD digital camera at a frame rate of 18 fps. For exploring the influence of pH values on motion behaviors, the H_2_O_2_ concentration of the whole system was diluted to 2.5 wt%, and their pH values were adjusted to 4.4, 5.0, 5.6, 6.2, 6.8 and 7.4 using 20 mM disodium hydrogen phosphate-citrate buffer solution. For studying the influence of H_2_O_2_ concentrations on motion behaviors, concentrations of 0.5 wt%, 1 wt%, 2.5 wt%, 5 wt% and 10 wt% were adopted, respectively. For the spontaneous chemotactic experiments, agar (0.5 cm × 0.5 cm) was put in 2.5 wt% H_2_O_2_ solution (pH 6.8) for 12 h at room temperature in advance, then taken out and put on a hydrophobic glass substrate. Subsequently, 2.5 wt% H_2_O_2_ solution (pH 7.4), containing the FCNMs, was dropped on the other side. For the magnetic manipulation experiments for the FCNMs, 2.5 wt% H_2_O_2_ solution (pH 6.8), containing the FCNMs, was dropped on a hydrophobic glass substrate, and the direction of its motion was manipulated by moving an external magnet.

### 2.7. Characterization

The scanning electron microscope (SEM) images were taken with a Hitachi S-4800 microscope (Tokyo, Japan). The transmission electron microscopy (TEM) and energy-dispersive X-ray (EDX) images were taken with a FEI F200 microscopy (Hillsboro, OR, USA). The powder X-ray diffraction (XRD) patterns were obtained with a Empyrean diffractometer (Almelo, Holland). Magnetic properties were analyzed with a Lake Shore 7400 vibrating sample magnetometer (VSM, Westerville, OH, USA). The thermal properties were evaluated with a STA 449F3 thermogravimeter (TG, Selb, Germany). The dynamic light scattering (DLS) measurements were proceeded with a NanoBrook 90Plus Zeta (New York, NY, USA). The motion videos and images were recorded with a Leica DM 3000B optical microscope (Wetzlar, Germany). 

### 2.8. Numerical Simulation

Our numerical model was implemented in COMSOL Multiphysics (version 5.4a) in a 2D axisymmetric configuration to reduce the computational cost. The primary model parameters are shown in Appendix A and Appendix A in the Appendix A. The chemical reaction engineering and dilute chemical species transport module were employed. Fe_3_O_4_ NPs were presumed to distribute uniformly in the flask bottom with a constant concentration of 7.14 × 10^−5^ mol·m^−2^. When the FCNMs moved in an acid condition (pH 4.4), chemical reactions: $\equiv Fe^{2+}+H_{2}O_{2}\to \equiv Fe^{3+}+OH^{-}+•OH\;$ and $OH^{-}+C_{6}H_{8}O_{7}⇆C_{6}H_{7}O_{7}^{-}+H_{2}O\;$ occurred in the confined cavity, which were calculated by a chemical reaction engineering model with a rate constant of 65 M^−1^·s^−1^ and 1.1 × 10^10^ M^−1^·s^−1^, respectively, and the governing equation was as follows:(1)$$\frac{dc_{i}}{dt}=R_{i}+R_{ads,i}\frac{A_{r}}{V_{r}}$$

Here, for a species i, $c_{i}$ is the concentration, $R_{i}$ is the rate of the species, $R_{ads,i}$ is the surface reaction area, and $V_{r}$ is the reactor volume. Transport of diluted species is used to compute the concentration field with the governing equation: (2)$$\nabla J_{i}=u\nabla c_{i}-D_{i}\nabla^{2}c_{i}$$where $J_{i}$ and $c_{i}$ are the ionic flux and ionic concentration for species *i*, respectively, u is the flow velocity, D is the ionic diffusion coefficient. We noted that diluted species distributions were affected by flow field and chemical potential field. Chemical reaction: $OH^{-}+C_{6}H_{8}O_{7}⇆C_{6}H_{7}O_{7}^{-}+H_{2}O$ took place in fluid bulk. Rather, the reaction: $\equiv Fe^{2+}+H_{2}O_{2}\to \equiv Fe^{3+}+OH^{-}+•OH$ took place on the catalytic surface, and the boundary fluxes at the catalytic surfaces thus became:(3)$$n\cdot \left(-D_{i}\nabla c_{i}\right)=R_{ads,i}$$

The FCNM was driven by both chemical potential fields generated by concentrations of ionic species and non-ionic species, respectively. As the FCNM was fixed in the model, we set the FCNM surface as the slip boundary condition, and combined chemiosmotic flow, with a slip velocity, as:(4)$$\begin{array}{c} u=u_{ionic}+u_{non-ionic}\\ =\left(I-nn\right)\left(b_{p\_ionic}\cdot \sum \left(\nabla c_{ions}\right)+b_{p\_non-ionic}\cdot \sum \left(\nabla c_{non-ions}\right)\right) \end{array}$$where $b_{p\_ionic}$ and $b_{p\_non-ionic}$ are the surface mobility of the FCNM, caused by concentrations of ionic species and non-ionic species. On account of solvation of ions, $b_{p\_ionic}$ and $b_{p\_non-ionic}$ have opposite signs, which means opposite direction of flow. And the quantity (I−nn) defines the concentration gradients, where I denotes the second-order unit tensor. Fluid field around the FCNM was solved by Stokes equation:(5)$$\nabla p=\eta \nabla^{2}u,\;\;\nabla \cdot u=0$$where p is the pressure, η is dynamic viscosity of water.

When the FCNMs moved in a near alkaline condition (pH 7.4), the chemical potential field caused by $\nabla c_{O_{2}}$ was considered the main source of driving force. The boundary conditions at the outer edge of the simulation domain were chosen to represent the bulk: constant concentrations of species, and a no stress boundary condition for fluid. At the FCNMs’ surfaces, we prescribed a uniform charge density. We solved for transient state for the FCNMs, which shows the system reached a stable state after 1 ms, hence, results in the discussion all are at 1 ms.

## 3. Results and Discussion

### 3.1. Implementation Strategy of the pH-Responsive FCNMs

The self-adaptive physiological pH-responsive FCNMs were realized by virtue of the dual-enzyme catalytic reactions of Fe_3_O_4_ NPs for H_2_O_2_ decomposition inside the confined cavity of flask-like structures (Figure 1). When Fe_3_O_4_ NPs encountered the catalyzed substrate of H_2_O_2_, the following reactions occurred:(6)$$\equiv Fe^{2+}+H_{2}O_{2}\to \equiv Fe^{3+}+OH^{-}+•OH$$
(7)$$\equiv Fe^{3+}+H_{2}O_{2}\to \equiv Fe^{2+}+HO_{2}•/O_{2}^{-}+H^{+}\;\;$$
(8)$$•OH+HO_{2}•/O_{2}^{-}\to H_{2}O+O_{2}$$
(9)$$OH^{-}+C_{6}H_{8}O_{7}⇆C_{6}H_{7}O_{7}^{-}+H_{2}O\;\;\;\;\;$$

At a lower pH (e.g., pH < 5.6 in Figure 1), Equation (6) dominated with a rate constant of 65 M^−1^·s^−1^ to produce •OH, and Equations (7) and (8) were inhibited because of the acidity of the solution, indicating its peroxidase-like activity. To eliminate the effect of ion by-products on the pH value, all motions of the as-prepared FCNMs were observed in a 20 mM buffer solution. At this time, the FCNMs were driven by the co-effect of C_6_H_7_O_7_^−^ and •OH diffusiophoresis in the opposite directions. With increasing pH to near neutral or alkaline, Equation (7) was the rate-determining step (k = 2.5 × 10^−3^ M^−1^·s^−1^) was promoted, and O_2_ was produced instead of •OH (Equation (8)). This could lead to the driving force having to convert into O_2_ diffusiopheresis. Therefore, with increasing pH in the acidic environment (pH < 5.6), the decrease of peroxidase-like activity could reduce the concentrations of C_6_H_7_O_7_^−^ and •OH, and, thus, decrease the motion velocity of the FCNM. On further increasing pH from 5.6 to 6.8, the diffusiophoresis of C_6_H_7_O_7_^−^ was weakened until it disappeared, because the occurrence of Equations (7) and (8) decreased the concentration of C_6_H_7_O_7_^−^ via producing H^+^ to neutralize OH^−^, and simultaneously generated O_2_ for driving the FCNMs. This resulted in increase of the moving velocity. When pH > 6.8, the catalase-like activity of Fe_3_O_4_ dominated instead of peroxidase-like activity and O_2_ diffusiophoresis acted as the main driving force. Due to the relatively low-rate constant of Equation (7), the concentration of O_2_ was lower in the system and the velocity decrease might have also been due to the diffusion restriction of the flask-like structures. As a result, the as-prepared FCNMs displayed self-adaptive motion behaviors following the pH changes in the system.

### 3.2. Preparation and Characterization of the FCNMs

The FCNMs were prepared by loading Fe_3_O_4_@DMSA NPs in flask-like carbonaceous carriers (FCCs), as schematically illustrated in Figure 2a. The FCCs were synthesized by a soft-template-assisted hydrothermal method with a relatively uniform size of 600 nm, a cavity of ~280 nm and a flask mouth of ~140 nm (Appendix A). Their surfaces were rich in hydroxyl and carboxyl groups giving rise to a lower surface potential (−57 mV). Then, using the thermal decomposition method, different Fe_3_O_4_@OA NPs were synthesized by regulating experimental parameters, including aging temperature (Appendix A), aging time (Appendix A) and raw material molar ratios (Appendix A). The results indicated that the as-synthesized nanoparticles were magnetite (Fe_3_O_4_; JCPDS no. 75-0033) with a cubic inverse spinel structure, and simultaneously, it could also be found that the particle size and saturation magnetization increased with increasing aging temperature, time or decreasing the amount of oleyl alcohol. Subsequently, the nanoparticles with the highest catalytic activity (Appendix A and Appendix A) were modified by DMSA to achieve transformation from hydrophobicity to hydrophilicity (Appendix A). Next, the morphology and structure of the as-prepared FCNMs were characterized by SEM (Figure 2b) and TEM (insert), showing smooth outer surfaces and large inner cavities with particles. The EDX images shown in Figure 2c clearly confirmed the loading of the Fe_3_O_4_ NPs in the cavity of the FCCs. The XRD patterns (Figure 2d) further verified that the FCNMs were composed of amorphous FCCs and Fe_3_O_4_ NPs. The loading amount of Fe_3_O_4_ NPs was evaluated to be 10.22% by TG (Figure 2e). The saturation magnetization of the FCNMs was 7.4 emu·g^−1^, which was much lower than that of Fe_3_O_4_ NPs (40.9 emu·g^−1^, Figure 2f), with no hysteresis, demonstrating their superparamagnetic features favoring magnetic field manipulation. Overall, Fe_3_O_4_-based flask-like carbonaceous nanomotors were successfully constructed.

### 3.3. Motion Behaviors of the FCNMs at Different pH Values

By virtue of the dual enzyme-like activities of Fe_3_O_4_ NPs that can react with H_2_O_2_ to produce •OH in acidic environment (peroxidase-like activity) and O_2_ in near neutral and alkaline environments (catalase-like activity), it is conceivable that the as-prepared FCNMs could exhibit pH-dependent motion behaviors. In term of this, we explored the motion behaviors of the FCNMs at 2.5 wt% H_2_O_2_ aqueous solution in physiological pH values ranging from 4.4 to 7.4 with a gradient of 0.6. Since H_2_O_2_ decomposition usually causes the pH change in solution, the motion behaviors were conducted in a 20 mM buffer solution and observed using an inverted optical microscope. In each case, at least 15 FCNMs were analyzed. Typical tracking trajectories of the FCNMs with, or without, H_2_O_2_ “fuel” are presented in Figure 3a. In the absence of H_2_O_2_, the FCNMs performed Brownian motion only (Appendix A), while those with the presence of H_2_O_2_ exhibited active self-propulsion, indicating that the driving force for the motion derived from the reaction between FCNMs and H_2_O_2_. The motion behaviors of the FCNMs exhibited a non-linear change with increasing pH values, which was further verified by the mean-square-displacement (MSD) versus the time interval (Δ*t*) curves, according to the extracted 2D coordinates along the trajectories (Figure 3b). Without adding H_2_O_2_, the linear relationship between the MSD and Δ*t* indicated a typical Brownian diffusive motion. The parabolic MSD curves, when using H_2_O_2_ as the fuel, suggested directional self-propulsion of the FCNMs. In order to clarity the relationship, we calculated the average velocities (Figure 3c) and diffusion coefficients (Figure 3d) of the FCNMs versus the pH values, which suggested their self-adaptive pH responsiveness. The velocities of the FCNMs decreased with increasing pH values from 4.4 to 5.6, and then rose until pH value was up to 6.8 with a velocity of 7.42 ± 1.0 μm/s (≈12.4 body lengths·s^−1^), after which the velocities subsequently dropped when pH was 7.4 (Appendix A). A similar dependence could also be found in diffusion coefficients of the FCNMs at different pHs calculated by the equation of $MSD=4D_{\mathrm{eff}}\Delta t$. This confirmed that the as-prepared FCNMs could self-regulate their velocities, by sensing pH values in the physiological range, which provided a possibility for their self-adaptive movement toward a specific pH.

### 3.4. Experimental Verification and Simulation Analysis for the pH-Responsive Mechanisms

In order to figure out why the FCNMs could exhibit pH-responsive motion behavior, we conducted a deep exploration for propulsion mechanisms through experimental verification and simulation analysis. The relationship between the dual enzyme-like activities of Fe_3_O_4_ NPs and environmental pH was studied firstly. The peroxidase-like activity of Fe_3_O_4_ NPs was tested by catalyzing the oxidation of substrate 3,3′,5,5′-tetramethylbenzidine (TMB) in the presence of H_2_O_2_, which was accompanied by color change to blue and appeared as obvious UV-Vis absorption at 650 nm (Appendix A). The results showed that the peroxidase-like activity increased with increasing H_2_O_2_ concentrations and reached the maximum at 2.5 wt%, whereas it decreased with increasing the pH values till at pH 7.4 there was no •OH production. Instead, when pH > 5.6, O_2_ generation could be detected by monitoring dissolved oxygen in the reaction system, indicating the catalase-like activity of the as-synthesized Fe_3_O_4_ NPs. Then, we focused on verifying whether this law held true once the Fe_3_O_4_ NPs were encapsulated in a confined space to form a motor. The changes of the light absorption value at 650 nm in the first 1000 s of the reaction system were plotted in Figure 4a. It could be found that as the pH increased, the slope values in the linear range decreased and became zero once the pH exceeded 6.8, and, at the same time, the amounts of O_2_ generation increased (Figure 4b). This suggested that, with increase of the pH values, Equations (7) and (8) could be activated to expend •OH, resulting in the decrease until it disappeared in the system. Further increasing the pH values, O_2_, instead of •OH, became the main product. Therefore, in terms of the results it was concluded that with increasing pH values from 4.4 to 7.4, the main products of the system experienced three stages: only •OH generated, •OH and O_2_ coexisted, and O_2_ dominated. This undoubtedly led to the transformation of propulsion mechanisms.

To further elucidate the self-adaptive pH-responsive motion behaviors of the FCNMs, we simulated the flow field distributions at different pH values, respectively, using COMSOL multiphysics based on a 2D model. On the basis of Figure 4a,b, only Equation (6) occurred at a lower pH (e.g., pH < 5.6), •OH was the main product accompanied by the generation of OH^−^ (Appendix A). In order to exclude the effect of OH^−^ on the pH value, Equation (9) occurred and the ionic product C_6_H_7_O_7_^−^ was generated. Therefore, the FCNMs in this case were co-driven by ionic and non-ionic diffusiophoresis. As shown in Figure 4c, the ionic concentration gradient formed at the opening, while the non-ionic concentration gradient was at the inner neck in an opposite direction. Therefore, the motion direction of the FCNMs depended on their relative magnitude. The simulated result showed that non-ionic diffusiophoresis played a dominant role in the current system, which drove the motor moving toward the flask mouth direction (Figure 4d). Limited by the resolution of the used optical microscope, the motion direction of the as-prepared FCNMs could not be distinctly resolved. Therefore, we treated much larger FCCs with a similar structure (~1 μm) to fabricate micromotors to experimentally verify the motion direction (Appendix A and Appendix A), which was in good agreement with the simulated result, and suggested the dominant effect of non-ionic diffusiphoresis. The reduced motion velocity in acidic pH from 4.4 to 5.6 could be attributed to the reduction of peroxidase-like activity of Fe_3_O_4_ NPs. When the pH value was up to 5.6, Equations (7) and (8) were activated to not only consume •OH groups but also to produce H^+^, which was evidenced by the appearance of dissolved O_2_ in the system. At this time, OH^−^ ions produced from Equation (6) were partly neutralized by H^+^ ions giving rise to the reduction of ion concentrations, and thus weakening the ionic diffusiophoresis. Additionally, O_2_ generation could, to some extent, increase the non-ionic dissusiophoresis, and, thus, accelerate the FCNMs. With the disappearance of ionic diffusiophoresis in the system and the complete conversion of •OH groups, in terms of Equation (8), the movement velocity of the FCNMs reached a maximum at pH 6.8. The subsequently reduction of the velocity in near alkaline range might derive from the relatively low O_2_ concentration gradient (Appendix A) and the diffusion restriction of the flask-like structures. From the point of view of the relative strength of the flow fields, we could also find that the non-ionic diffusiophoresis from •OH was much stronger than that of O_2_, which was in keeping with their concentration gradient distribution and the experimental motion behavior. 

### 3.5. Motion Behaviors of the FCNMs at Different H_2_O_2_ Concentrations

Figure 5 presents the effect of H_2_O_2_ concentrations on the motion behaviors of the as-prepared FCNMs. In each case, at least 15 FCNMs were analyzed. The results display that the FCNMs at 0 wt% H_2_O_2_ performed Brownian motion only, while those with the presence of H_2_O_2_ exhibited active self-propulsion with various trajectories, dependent on the H_2_O_2_ concentrations (Figure 5a). Parabolic MSD curves in Figure 5b show increasing slope values in the linear range until the H_2_O_2_ concentration was up to 2.5 wt%, also suggesting a transition from Brownian diffusive motion to self-propulsion. We also calculated the average velocities and diffusion coefficients of the FCNMs versus the H_2_O_2_ concentrations. The mobility of the FCNMs continued to increase with increasing H_2_O_2_ concentrations at the beginning, and dropped obviously when the H_2_O_2_ concentration was higher than 2.5 wt% (Figure 5c,d). It is suggested that more H_2_O_2_ might increase the ionic strength in the system, which had a negative effect on the motor movement. This phenomenon is consistent with the description of Sen et al., where it was proposed that when the total ionic strength of the system increases, there is a great negative effect on the motion capability of the motors driven by ionic diffusiophoresis [34]. The typical video of the FCNM swimming in 2.5 wt% H_2_O_2_ aqueous solution is provided in Video S5 with a maximum average velocity of 7.01 ± 0.59 μm/s (≈11.7 body lengths·s^−1^). 

### 3.6. Chemotactic Motion and Magnetic Responsiveness of the FCNMs

Thanks to the unique pH-responsive motion behaviors of FCNMs in the physiological range, they can display a self-adaptive chemotaxis motion toward a specific pH where they exhibit a maximum velocity. As shown in Figure 6a, a left-to-right gradually decreasing pH gradient was constructed by using a saturated agar filled with fuel solution at pH 6.8, which was placed on the other side of the mixed solution at pH 7.4 containing the FCNMs. In the presence of 2.5 wt% H_2_O_2_, the tracking trajectories of the 10 FCNMs were observed, and are depicted in Figure 6a. They all exhibited chemotactic behavior from pH 7.4 to 6.8 (Appendix A). Moreover, acceleration phenomena in the process of getting closer to the agar were observed for the 10 FCNMs by calculating their real-time velocities (Figure 6b). The average velocities of the corresponding FCNMs in the first 10 s and the last 10 s were also calculated in Appendix A. To avoid any accidental effect, we cleverly set up an opposite pH gradient to observe the motion direction of the FCNMs positioned on the side of the mixed solution at pH 6.8. As expected, the FCNMs were far away from the saturated agar filled with fuel solution at pH 7.4 (Appendix A and Appendix A). Therefore, the results provided a direction control strategy for the as-prepared FCNMs, which was capable of actively targeting a specific area with a unique pH value. Besides this, on-demand direction manipulation could also be achieved by external magnetic field because of the inherent superparamagnetism of Fe_3_O_4_ NPs (Figure 6c). Facilely magnetic separation and easy re-dispersion features also endowed the as-prepared FCNMs with excellent recycling characters (Figure 6d). 

## 4. Conclusions

In this work, we have demonstrated flask-like carbonaceous nanomotors (FCNMs) exhibiting self-adaptiveness toward a specifically physiological pH on the basis of the dual enzyme-like activities of magnetic Fe_3_O_4_ NPs. In the process of increasing pH in the physiological range from 4.4 to 7.4, the peroxidase-like activity for Fe_3_O_4_ NPs gradually wears off, accompanied by enhanced catalase-like activity, resulting in a transformation of the propulsion mechanisms from non-ionic and ionic diffusiophoresis to single non-ionic diffusiophoresis, which ultimately feeds back to a non-linear responsiveness upon pH. This endows the FCNMs with a unique chemotaxis towards a slightly acidic region (pH 6.8) that is a specific pH for generally physiological focal sites, such as tumor microenvironments. These results provide a design strategy for MNMs exhibiting a self-adaptive physiological pH-responsive motion, and the characteristic pH taxis, as well as the magnetic control, also offer a larger possibility in future biotechnological applications, especially for actively targeted tumor therapy.

## Figures and Tables

**Figure 1 nanomaterials-12-02049-f001:**
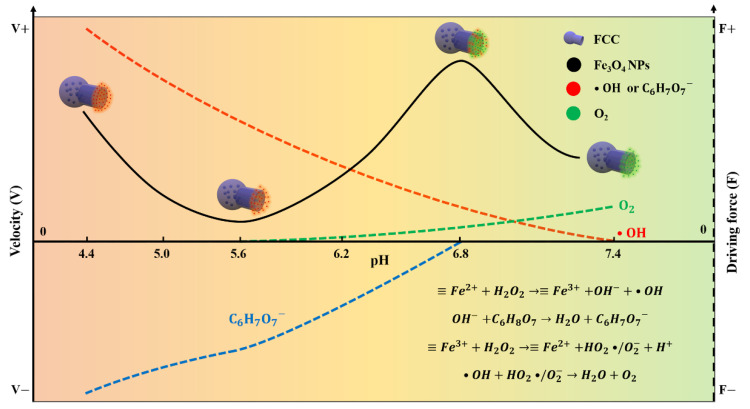
Schematic of the self-adaptive pH-responsive motion behaviors of the as-prepared FCNMs based on the pH-dependent catalytic activities of Fe_3_O_4_ NPs. The velocity and driving force toward the flask mouth direction were set to be positive (V+ and F+). At a lower pH value, the FCNMs were driven by the co-effect of C_6_H_7_O_7_- and •OH diffusiophoresis in the opposite directions. With the reduction of ionic diffusiophoresis and the addition of O_2_ diffusiophoresis, the motion velocity increased in the weak acid environment. In the near alkaline range, O_2_ diffusiophoresis provided the driving force for the FCNMs showing a reduction of the velocity.

**Figure 2 nanomaterials-12-02049-f002:**
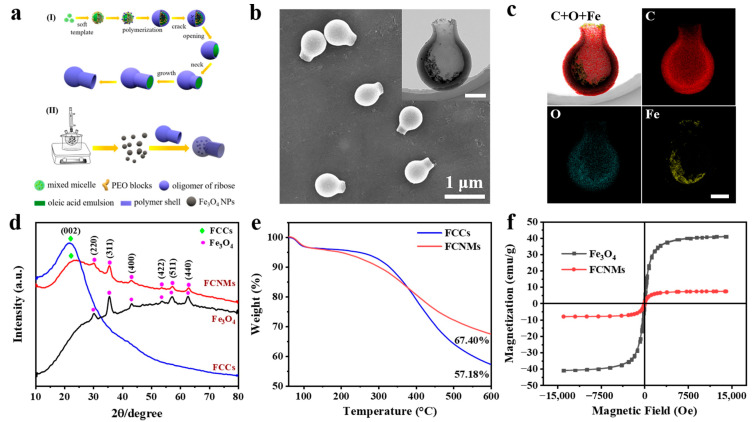
Fabrication and characterizations of the FCNMs. (**a**) Schematic illustration of the experimental procedures. (**b**) SEM image of as-prepared FCNMs (The inset is the TEM image of a typical FCNM. Scale bars: 200 nm.). (**c**) Corresponding EDX images of the distribution of C, O and Fe. Scale bars: 200 nm. (**d**) XRD patterns of the Fe_3_O_4_ NPs, FCCs and FCNMs. (**e**) TG curves of the FCCs and FCNMs. (**f**) Magnetic hysteresis loops of the Fe_3_O_4_ NPs and FCNMs.

**Figure 3 nanomaterials-12-02049-f003:**
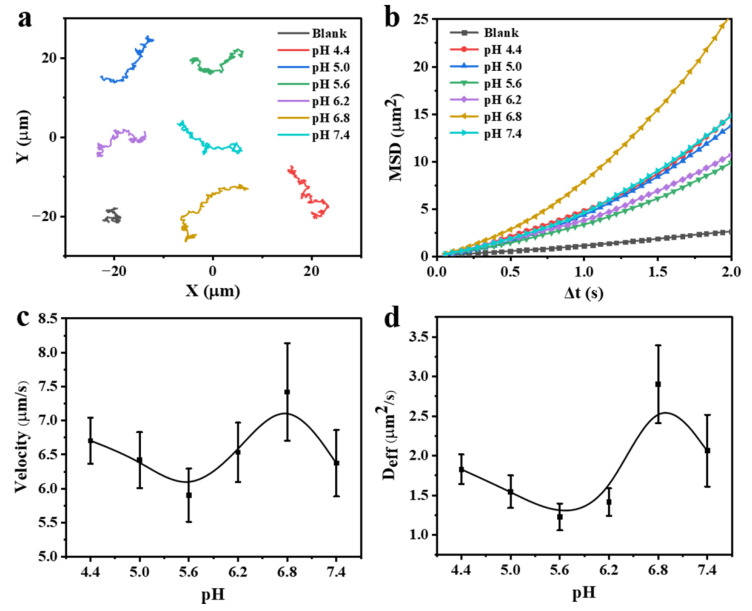
Motion analysis of the FCNMs at different pH values in 2.5 wt% H_2_O_2_ aqueous solution. (**a**) Track trajectories of the autonomous motion of the FCNMs in 10 s. (**b**) MSD curves versus Δ*t*, (**c**) corresponding average velocities, and (**d**) average diffusion coefficients of the FCNMs.

**Figure 4 nanomaterials-12-02049-f004:**
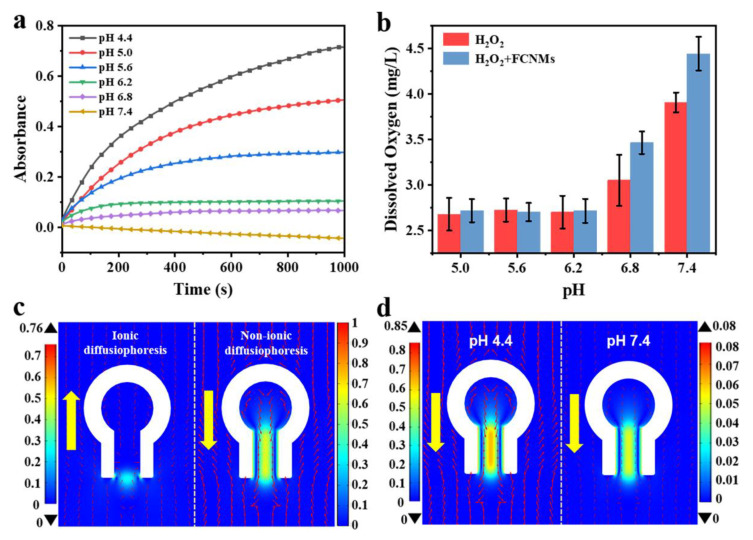
Verifications of propulsion mechanisms of the FCNMs through experiments and numerical simulations. (**a**) UV-Vis absorption-time course curves at 650 nm of the TMB-H_2_O_2_ system with the FCNMs at different pH values. (**b**) Dissolved oxygen analysis of the H_2_O_2_ system with or without the FCNMs at different pH values for 10 min. (**c**) Flow fields caused by ionic and non-ionic concentration gradient around the FCNMs at pH 4.4, respectively, where the small red arrows refer to the relative strength of flow velocity and the large yellow arrows refer to the direction of motion. (**d**) Total flow fields around the FCNMs at pH 4.4 and 7.4, respectively.

**Figure 5 nanomaterials-12-02049-f005:**
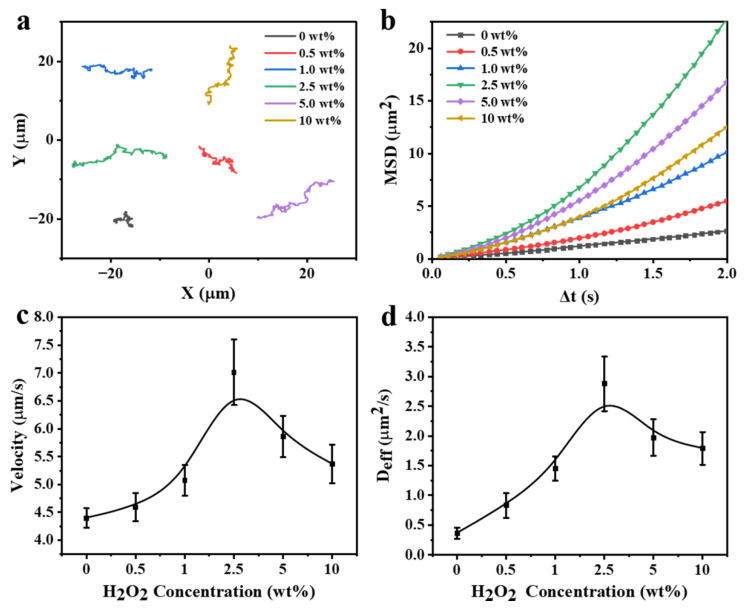
Motion analysis of the FCNMs at different H_2_O_2_ concentrations. (**a**) Track trajectories of the autonomous motion of the FCNMs in 10 s. (**b**) MSD curves versus Δ*t*, (**c**) corresponding average velocities, and (**d**) average diffusion coefficients at different H_2_O_2_ concentrations.

**Figure 6 nanomaterials-12-02049-f006:**
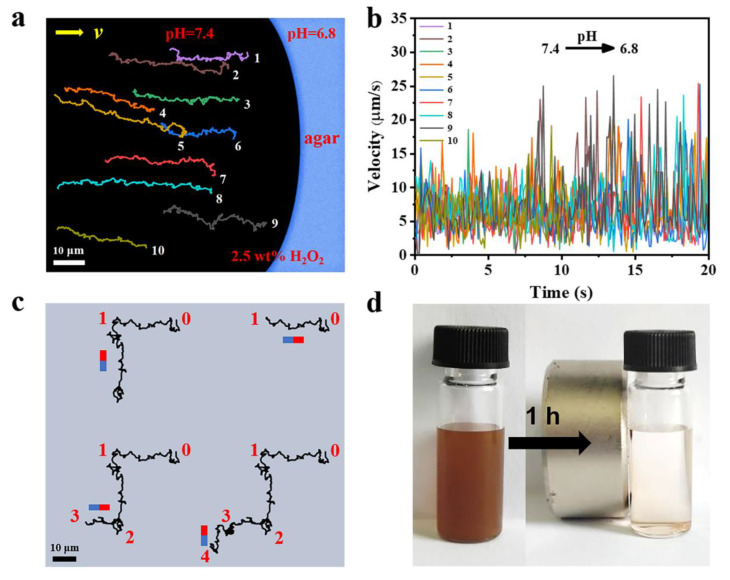
Chemotactic motion of the as-prepared FCNMs from pH 7.4 to 6.8 and magnetic responsiveness of the FCNMs. (**a**) The chemotactic track trajectories of the FCNMs in two pH environments in 30 s. (**b**) Real-time velocities for different FCNMs in 20 s. (**c**) Motion trajectories of a typical FCNM in 2.5 wt% H_2_O_2_ aqueous solution at pH 6.8 under magnetic manipulation. (**d**) Photograph of the FCNMs dispersed in aqueous solution and their magnetic separation for about 1 h.

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
