# Peer review of "Self-Adaptive Flask-like Nanomotors Based on Fe3O4 Nanoparticles to a Physiological pH"

_nanomaterials, 2022, doi:10.3390/nano12122049_

Round 1

Reviewer 1 Report

This version does not look worthy and cannot be recommended for publication in this form and at least needs some clarification and further revision.

  1. the abstract is too specific and immediately requires deepening into the text, otherwise it remains clear only to the authors.
  2. Since the Fe3O4 oxide is the main object of this work, a more detailed review as a short paragraph is also necessary! Give more motivational explanations why Fe3O4 and not Fe2O3. More information about size and preparation methods will be useful as well as their possible influence on results. For a wider range of readers, more information on Fe3O4 preparation methods would be desirable. See for examples, some of them published this year in MDPI journals,

Serga, V.et al . Impact of Gadolinium on the Structure and Magnetic Properties of Nanocrystalline Powders of Iron Oxides Produced by the Extraction-Pyrolytic Method. Materials 2020, 13, 4147.

Li, Y.; et al. Superparamagnetic α-Fe2O3/Fe3O4 Heterogeneous Nanoparticles with Enhanced Biocompatibility. Nanomaterials 2021, 11, 834

  1. I would like to see before the conclusions what new data on nanoparticles were obtained and which could be put into a type of overview table.
  2. Figure 4a, namely its independent explanation does not make it possible to see for which spectral region the measurements were made.

Author Response

Dear professor, please check the attachment.

Reviewer 2 Report

Authors shows an interesting system to study pH-taxis through the use of flask nanomotors that for sure it will be attractive for Nanomaterials readers. The results are consistent, and the materials were well characterized. However, I have some comments before this work can be published in Nanomaterials.

Comment 1: State of art on micromotors and nanomotors topic is incomplete. Add a text including the next references related to nano/micromotor for biomedical applications:

  • C. Horetlao, et al. Swarming behavior and in vivo monitoring of enzymatic nanomotors within the bladder. Science Robotics 6 (2021), eabd2823
  • Vilela, et al. Drug-Free Enzyme-Based Bactericidal Nanomotors against Pathogenic Bacteria. ACS applied materials & interfaces 13 (2021), 14964-14973
  • A Llopis-Lorente, et al. Enzyme-powered gated mesoporous silica nanomotors for on-command intracellular payload delivery. ACS nano 13 (2019), 12171-12183
  • Motion control of urea-powered biocompatible hollow microcapsules
  • X Ma, X Wang, K Hahn, S Sánchez. ACS Nano 10 (2016), 3597-3605

There are some works that have reported nanobottle synthesis and nanoflask based-motors such as:

  • Yao, et al. Janus-like boronate affinity magnetic molecularly imprinted nanobottles for specific adsorption and fast separation of luteolin. Chemical Engineering Journal
  • (2019) 436-444.
  • Qiu, et al. Nanobottles for Controlled Release and Drug Delivery. Adv Healthc Mater 10 (2021) e2000587.
  • Zhou, et al. Autonomous Motion of Bubble-Powered Carbonaceous Nanoflask Motors. Langmuir 2020, 36, 25, 7039–7045.
  • Surface Wettability-Directed Propulsion of Glucose-Powered Nanoflask Motors. ACS Nano 2019, 13, 11, 12758–12766.

Comment 2: From figure 3c we cannot observed a significant speed change depending on the concentration. How many particles at the same conditions have the authors used? Similar is showed in figure 5. If n is bigger, the error bar will be reduced, and it will be possible to observe a significant different. Thus, author should carry out more analysis of the speed.

Comment 3: The authors envision the use of those systems for tumor therapy..however, the peroxide concentrations used are toxic. How this system can be used for biomedical applications?

Author Response

(The authors gave the same response as above.)

Reviewer 3 Report

Micro/nanomotors, that exhibit great potential in abundant applications such as biomimicry, biomedicine and environmental fields, achieve much progress in the past two decades. Considering the perquisite of biocompatibility towards clinical applications, enzyme-powered nanomotors provide nice power engines due to their high catalytic ability. However, on account of the intrinsic denature property, there is an urgent requirement to explore other candidates with enzyme-like property, but possessing high stability. In this end, nanozymes could be good candidates. To further investigate and explore the advantages of nanozymes-powered nanomotors, Guan et al. designed and constructed a flask-like carbonaceous nanomotors (FCNMs) encapsulated with Fe3O4 nanozymes, which showed an interesting pH dependent self-adaptive motility behavior. In this work, the authors clearly demonstrated the pH responsiveness of Fe3O4 nanozymes, in the meanwhile, wisely proved the motion mechanisms. The story of this work is very concise and clear, with enough data in the paper and supporting information, therefore making the reviewer believe this nice work could be accepted for publication.

Some minor points could be taking into consideration:

1, In the first paragraph of the introduction, some enzyme powered nanomotors could be cited as background, such as lipase-powered nanomotors or protocells published in Angew. Chem. 2019, 2020, by Wang et al;

2, some of the error bars in Figure 3 and 5 should be repeated, considering the current version is in relatively large range;

3, Figure 4 was embedded into the paragraph, which needs to be moved out.

Author Response

(The authors gave the same response as above.)

Round 2

Reviewer 1 Report

the authors responded constructively to the questions, the article can be recommended for publication